# The Dynamics of *Lepus granatensis* and *Oryctolagus cuniculus* in a Mediterranean Agrarian Area: Are Hares Segregating from Rabbit Habitats after Disease Impact?

**DOI:** 10.3390/ani12111351

**Published:** 2022-05-25

**Authors:** José Prenda, Eduardo J. Rodríguez-Rodríguez, Juan J. Negro, Juan Manuel Muñoz-Pichardo

**Affiliations:** 1Department of Integrative Sciences, University of Huelva, 21071 Huelva, Spain; eduardo.rodriguez@dci.uhu.es; 2Department of Evolutive Ecology, Estación Biológica de Doñana-CSIC, Avenue Americo Vespucio, 26, 41092 Seville, Spain; negro@ebd.csic.es; 3Department of Statistics and Operative Research, University of Seville, 41012 Seville, Spain; juanm@us.es

**Keywords:** Lagomorpha, anthropized landscape, spatial distribution, myxomatosis, rabbit, hare

## Abstract

**Simple Summary:**

Hares (*Lepus* sp.) and rabbits (*Oryctolagus cuniculus*) are the two genera of lagomorph present in the Iberian Peninsula, and both are an essential element of the trophic chain in this region. The myxomatosis disease has produced mortalities in rabbits for decades, but this disease has recently jumped to the most widely distributed hare species in the Peninsula, the Iberian hare (*Lepus granatensis*). In this work, we analyzed the effects of this outbreak on the populations of both species in a Mediterranean agricultural landscape, and provide information about spatial and temporal patterns prior to and after the epidemic.

**Abstract:**

The genera *Oryctolagus* and *Lepus* (order Lagomorpha) are essential elements in the trophic chain in the Iberian Peninsula, being the main prey of many predators, including some highly endangered predators such as the Iberian lynx (*Lynx pardinus*). Myxomatosis, a disease producing tumorations in conjunctive tissues, and produced by the *Myxoma Virus*, has caused mass mortalities in rabbits (*Oryctolagus cuniculus*) for decades. Recently, the virus has jumped interspecifically from rabbits to hares, and this has created a depletion in hare populations, generating great concern. We analyzed the population dynamics and distribution of both lagomorph species in a Mediterranean agricultural area of the south of Spain since the 1990s with a combination of systematic and non-systematic data. The appearance of the outbreak in the Iberian hare (*Lepus granatenis*) in 2018 enabled us to undertake an opportunistic analysis of its effects on the spatial structure and assemblages, as well as on the niches of both species using PCA analyses and ordination techniques. Analysis of the mortality effect on daily and seasonal cycles was also conducted, and relations with the temporal dimension was tested using generalized lineal models (GLMs). In our results, in addition to population and temporal patterns, we could observe a restructuring in hare distribution after the mortality event, highlighting that prior to the outbreak, rabbit and hare populations were spatially differentiated, although with some overlaps and niche similarities. However, since the outbreak, hare populations have been excluded from rabbit areas, suggesting that in the absence of rabbits, the virus has more difficulties to infect hares. We also provide an overview of the effect of this population depletion on the ecological and socio-economic dimension of this region, pointing out the importance of this situation for the area.

## 1. Introduction

In the Iberian Peninsula, lagomorphs constitute a fundamental element in the trophic chain, especially in the case of the rabbit (*Oryctolagus cuniculus*) [1], a keystone species in Mediterranean ecosystems [2]. Some predators have evolved to be specialists of this species [3,4], while others, although not strict specialists, are extremely dependent on rabbit [5,6]. At the same time, the three Iberian species of hare (the European hare *Lepus europaeus*, mountain hare *Lepus castroviejoi,* and Iberian hare *Lepus granatensis*) are also relatively important in the diet of carnivore species and are also coveted by game hunters [7]. *L. granatensis* is the widest distributed hare species in the Peninsula, and the only one present in the south of Spain [8] and Portugal. Both the rabbit and Iberian hare are sympatric species over large parts of the Iberian Peninsula [7]. However, little is known about the way these species distribute spatio-temporally at the local scale and how this can drive potential interactions. Tapia et al. [9] did analyze the habitat use by both species but without any reference to actual or potential interactions affecting the observed patterns. Some studies on lagomorph interactions are available, especially concerning the European hare and rabbit, but in all cases, these studies have concerned areas where both species had been introduced [10] and none of these studies have reported any coherent tendency for the relationship between them. Inconclusive interaction patterns emerged depending on climatic, habitat, and physiographic conditions [10]. However, Barnes and Tapper [11], in a study of rabbits and hares in North Anglia, concluded that rabbit and hare numbers were associated between and within farms, although there was some evidence that hare numbers could be reduced on some sites where the rabbit density was extremely high. Nonetheless, these authors did not find any evidence for an inverse correlation between rabbits and hares that would explain why the disappearance of rabbits lead to an increase in hares. Additionally, Broekhuizen [12] reported that hares avoided rabbits. Clarifying these relationships are of major importance, both from an ecological and conservation point of view, but also for the significance of these species as game.

To study the long-term local patterns of Iberian hare and rabbit distribution in a highly anthropized agricultural area, on the outskirts of several midsize towns in southern Spain, we registered and geolocated all direct observations and indirect signs of their presence (droppings, footprints, and burrows). The area, focusing on the countryside around Carmona (Seville province, S. Spain), corresponds to the typical Mediterranean non-irrigated agricultural open land of mixed crops (cereals, sunflower, legumes, olive trees, etc.). Several years of lagomorph data had been collected when an epidemic outbreak occurred within the hare population, caused by the MYXV-tol virus [13] and ascribed to myxomatosis according to external symptoms and veterinary analyses [14,15]. Dead and ill hares with myxomatosis symptoms were initially collected in the study area by local hunters. These samples were subsequently analyzed by the veterinary services of the Junta de Andalucía, which confirmed this disease and kept track of it. This unexpected event, a quasi-experiment, is of major importance, given the information it can provide on the relationships and spatial dynamics of both lagomorph species, in addition to the interesting clues it offers for interpreting the completely novel phenomenon that this hare outbreak represents. At the same time, the consequences of the outbreak are of great concern, not only from an ecologic or conservationist point of view but also from economic one, as this area is nationally renowned for hare hunting, especially with greyhounds.

The conservation of lagomorphs, especially the Iberian hare, is affected by several factors which include natural predators, hunting pressure, habitat loss and fragmentation, use of pesticides, climatic change, roadkill, and recently, disease [16]. From the mid-20th century to date, the irruption of myxomatosis, a disease caused by the *myxoma virus* that produces conjunctive tumorations, has produced high mortality events, and synergically with the rabbit hemorrhagic disease (RHD), a catastrophic decline in populations of rabbits [17,18]. This has generated a severe conservation problem for several species of native Iberian predators, including some endemic species such as the endangered Iberian lynx (*Lynx pardinus*) and the vulnerable Iberian imperial eagle (*Aquila adalberti*) [19,20], the current conservation programs for which rabbit abundance is deemed critical. Rabbit myxomatosis is characterized by periodic outbreaks in the breeding season (spring and early summer), mainly affecting young animals, as most adults present antibodies [17]. Since 2018, myxomatosis has irrupted in the Iberian hare populations, generating massive mortalities [15]. The outbreak is unprecedented, although the disease has been reported previously in the European brown hare. This epizootic is now widely distributed all over the Iberian Peninsula [21,22], but its consequences on hare abundance, local distribution, and spatio-temporal relationships with rabbits and other species within the same trophic web remain unknown. Additionally, the decrease in hares throughout the Iberian Peninsula is not only an ecological problem but also of great socioeconomic concern, as the hunting of this prey is a traditional activity that maintains a strong economic chain [23], with trained dogs, game wardens, hunting societies, contests, and even a black economy in the form of illegal gambling. The considerable social concern generated in our study area has prompted hunting foundations, for example, to create a collaborative platform for monitoring the impact of myxomatosis on hares (*Mixolepus*. https://fundacionartemisan.com/investigacion/grupo-mixolepus-liebres/) (accessed on 10 February 2022).

The general goal of this study was to analyze the long-term spatio-temporal dynamics and habitat use of rabbit and hare populations. The sudden outbreak of myxomatosis in both lagomorphs within the study area provided an opportunity to investigate how the populations of two demographically different species were affected, and to quantify their interactive capacity, based on population and habitat interactions. To achieve this goal, we studied (a) the long-term interannual (1995–2020) population dynamics of hares and rabbits, focusing particularly on the myxomatosis outbreak, (b) their annual and daily cycles, and (c) their spatial relationships, habitat use, and niche assemblages, also considering the effect of the epidemic.

## 2. Methods

### 2.1. Study Area

The study area is an agricultural region near the city of Seville (S. Spain), in the middle of the lower Guadalquivir valley (Figure 1). This highly anthropized zone corresponds to the landscape unit of *Los Alcores y la Vega de Carmona* and include several mid-size towns and cities (Alcalá de Guadaira, Mairena del Alcor, El Viso del Alcor, Carmona, Arahal, Paradas and Marchena) totaling 192,248 inhabitants. The area is characterized by highly fertile agricultural soils, cultivated with typical Mediterranean crops, including cereals, legumes, sunflower, and patches of olive groves. Occasionally, there appear relicts of spontaneous Mediterranean vegetation. In the last decades of the 20th century, urban sprawl accelerated in the region, resulting in an increase in dispersed urban settlements, mainly in the northwestern part of the study area (Figure 1).

The relief is a flat alluvial plain (20–80 m.a.s.l.), locally called “La Vega”, disrupted only by a slightly sloped plateau (100–260 m.a.s.l.), or *Alcores*. While land use, and in consequence, the habitat in *La Vega*, is highly homogeneous agricultural land, the habitat in the *Alcores* is more heterogeneous with patches of bushland, urban areas, and crops (see Figure 1).

The climate is Mediterranean, with some oceanic effect, due to the proximity of the Atlantic Ocean and the dominant southwest winds. The annual mean temperature is 19.2 °C, with mild winters in which the minimum temperatures rarely drop below 5 °C, and extremely hot summers in which the mean maximum temperature exceeds 33 °C (July and August). Rainfall is concentrated in winter and spring, with an annual average of 550 mm (see Figure 1 for annual climatic data). Recurrent droughts followed by wet years are an important characteristic of the Mediterranean climate, and have a significant influence on species dynamics.

### 2.2. Lagomorph Censuses

Hare and rabbit counts were carried out over a period of 30 years, from 1990 to 2020. Data were collected following both systematic and non-systematic procedures. Non-systematic procedures were based on opportunistic observations made during field trips, usually by car, but also cycling or walking, with a mean distance of 61.3 km (range: 5.3–192.3 km; median: 62.2 km) over a total of 859 occasions or dates. On every occasion, between 1 and 45 census points were recorded (mean: 8.0 points; median: 5.0 points) to register the presence or abundance of Lagomorphs, based on direct observations or indirect signs (footprints, feces, or burrows) identified by the authors, drawing on extensive field experience. At every census point, the corresponding UTM coordinates and solar time were registered. The systematic approach was based on day-time line transect distance sampling which, according to Langbein et al. [24], is the census technique that provides the best reliability efficiency solution. Every transect was a fixed walk of 10 km in length near the town of Mairena del Alcor (Seville), traversed between December 2016 and March 2021 on 3984 dates. In this case, as previously, every direct observation or indirect sign of hares or rabbits was annotated, along with the UTM coordinates and the solar time.

For quantitative analysis, we considered only direct observations (indirect signs cannot be assigned a number of individuals, nor a precise date or time) in a two-fold manner (1) as frequency of occurrence (occurrences divided by the total number of census points) and (2) as relative density (individuals divided by total number of census points).

### 2.3. Data Analysis

Frequency distribution and density curves were constructed under several scenarios (25-year periods, annual cycle, and daily cycle) in order to compare the dynamics of both species and their changes through time. To avoid the effect of spurious relationships due to low sample size, we arbitrarily considered only years with *n* > 120 localities × date, or ideally, with *n* > 300 localities × date. We used the smooth density statistic to implement the adjustment between date and relative density (%).

In order to describe and visualize habitat use and spatial relationships between hares and rabbits, principal component analysis (PCA) was carried out on three alternative matrices: (1) coordinates (x, y, z) of every lagomorph observation made throughout the entire study period (1995–2020), (2) coordinates (x, y, z) of every sampling point during the intensive survey period (2016–2020), and (3) the same as in (2) but including the date of each register. With this procedure, it was possible to evaluate the relative distribution and the habitat overlap of both species within the study area and to track the spatial impact of the myxomatosis epidemic on them. Further, in order to test the effects of date and species on the daily activity time, we used a generalized linear model (Gaussian), with time (daily hour) as the dependent variable, and date and species (fixed factor) as the variables.

In addition to the strict spatial use of habitat, we also analyzed the niches of both species, so as to obtain a more detailed picture of the specific use of habitat elements. To compare these niche assemblages of both species prior to and following the myxomatosis outbreak, we used the package ECOSPAT [25], an ordination technique for the R environment [26] described in Di Cola et al. [27]. This package enabled us to conduct niche overlap (Schoener’s D through the niche PCA), equivalence (testing whether the niches are more equivalent than can be expected by chance), and similarity tests (testing whether niches are more similar than can be expected by chance) under the niche conservatism hypothesis (using the argument “=greater” in the similarity and equivalence test functions). The variables considered for these analyses were altitude, extracted from the digital elevation model of WorldClim2 [28], and the four raster layers included in the CORINE Land Cover of Andalusia [29,30]. The first level is a general classification of uses (agricultural, forestry, artificial surfaces, wetlands, and water bodies). The subsequent levels are more detailed hierarchal divisions of the first.

## 3. Results

### 3.1. Long-Term Population Tendencies of Hare and Rabbit

The relative density of Lagomorphs in the Carmona countryside varied throughout the 21 years of the study period, displaying an alternate sequence of minima and maxima in a saw-tooth pattern. Both hares and rabbits were similarly affected, although rabbits generally reached a higher relative density than hares (Figure 2). The temporal dynamics of hare and rabbit relative density did not show any global tendency over the study period (Spearman’s rank correlation (r_s_), *p* > 0.1), although both species were highly correlated (years > 120 localities × date: r_s_ = 0.49, *p* < 0.045, *n* = 18; years > 300 localities × date: r_s_ = 0.98, *p* < 0.0001, *n* = 10).

### 3.2. The Myxomatosis Outbreak

The year 2018 saw the beginning of a sharp decline in the relative density of Lagomorphs (January in the case of hares, February for rabbits), which in the case of rabbits ended in 2019, but in the case of hares continued through 2020 (Figure 2). This reduction was coincident with a myxomatosis outbreak, confirmed in the study area by the Sociedad Deportiva de Caza Arahalense (personal communication), and described elsewhere in Spain [14]. To date the start of this epidemic and its evolution, we analyzed our systematic censuses between December 2016 and December 2020, using both density and frequency of occurrence data. The overall pattern displayed by both species had two clearly differentiated parts, separated by a milestone in July 2018 (Figure 3). The hare and rabbit dynamics registered in 2017 seemed to be normal, with a maximum in summer that declined to a minimum in February 2018 for both species (Figure 3). However, from that date, in the case of rabbit, there was no subsequent recovery, especially in density. The hare displayed a secondary peak in April 2018 and then decreased to a minimum in July 2018 that lasted until the end of the study period (Figure 3). Rabbit and hare monthly patterns were highly correlated (Table 1). However, the frequency of rabbits experienced an earlier decline than hares (Figure 3a).

### 3.3. Annual Cycle

The long-term monthly dynamics of hare and rabbit in the Carmona countryside ran in parallel (Kendall tau = 0.53–0.76, *p* < 0.02, for the 1990–2020 period; Kendall tau = 0.56–0.74, *p* < 0.02, for the 2015–2020 period, *n* = 12) and both species peaked in summer (hare in July and rabbit in August) (Figure 4a,e), although there seemed to be a small time lag between the annual cycle of both species.

When this analysis was carried out for the period of intensive sampling only (16 December–20 December), distinguishing the periods before and after the myxomatosis outbreak in hares, the result was similar for rabbits, but quite different for hares (Figure 4g,h). Rabbits, independently of the epidemic, consistently displayed the same yearly pattern with maxima in August–September (Figure 4d,h). There was a high degree of correlation—all significant—between the four rabbit dynamics in Figure 4 (Kendall tau = 0.57–0.85, *p* < 0.05 in all cases). However, in the case of hares, the data for 18 July–20 December, during the epidemic, lacked any significant correlation with any other cases (Kendall tau = 0–0.23, *p* > 0.29 in all cases) (Figure 4). However, data for 16 December–18 June were still significantly correlated with hares sampled between 2015 and 2020 (Kendall tau = 0.52, *p* < 0.02). Thus, rabbits kept a rather constant cycle, independent of the time period and epidemic, while hares had a more variable cycle, highly altered during the outbreak of myxomatosis.

### 3.4. Daily Cycle

Hares and rabbits in the Carmona countryside apparently used the time during daylight non-selectively. The comparison between the frequency distributions of hourly records for each species with the available hour both for 1990–2019 and 2016–2020 periods did not display significant differences (Chi^2^ test, *p* > 0.05; Sup. matt. 1 and 2). This could be due to low sensitivity in the method employed, and the lack of consideration of any seasonal effect. To avoid the effects of seasonality, the hour at which each species was observed throughout the intensive sampling period is represented in Figure 5.

Detailed analysis of specific survey periods with a minimum sample size pointed to differing behavior between hares and rabbits (Figure 5b,c). The hares tended to be registered earlier than the rabbits, and the times they were registered tended to be less variable than in the case of rabbits, especially for the period from 17 June to 18 January. Species and date had a clear influence on time during the period from June 2017 to January 2018, whereas the relationship was less clear from March 2018 to July 2018 (see Glm model in Table 2).

### 3.5. Spatio-Temporal Relationships

In all PCAs (Table 3), Factor 1 arranged the sites in an E-W, N-S, altitudinal, and temporal gradient (Table 4). Factor 2 displayed a secondary, less statistically relevant geographical and temporal gradient. In PCA 1, however, there was no N-S or temporal gradient, only E-W and altitudinal. The altitudinal gradient separated the lower study sites, located on a flat herbaceous plain, from the higher sites, in a hilly woody area.

The spatial or spatio-temporal relationships between hares and rabbits in the space defined by Factors 1 and 2 from the three PCAs are shown in Figure 6. In all cases, rabbit distribution favored the north-west hills of the study area, while the hares were usually registered in the south-east plains. The position of hares and rabbits within the gradients defined by Factor 1 were statistically different in the three PCAs (*t*-tests, t > 8.6, *p* < 0.0001) (Figure 6). The coordinates of the centroids of hares and rabbits in the space defined by Factors 1 and 2 were different, defining a geographical distribution partially allopatric for both species (although rather overlapped). The amplitude of the space occupied by the rabbit was markedly higher than that of the hare. The former, more of a habitat generalist, had a wider distribution than the latter, and used the full availability of the gradients, while the hare was usually restricted to the lower portions of them. Hares mainly used the flat spaces of the valley with normally non-irrigated herbaceous crops (predominantly wheat, sunflower, and less frequently legumes), whereas rabbits were preferably distributed throughout areas close to or among the hills, with higher altitude and slopes, and which were generally less cultivated or covered with olive groves.

In addition, when considering the full study period, the hare was more abundant in flat herbaceous farm land (Pearson correlation with both Factor 1 and 2 from PCA 1: r = 0.30, *p* < 0.0001 and r = 0.17, *p* < 0.04, *n* = 152, respectively).

The abundance of rabbits significantly decreased throughout the study period, considering both the full time span and only the intensive survey (r = −0.21, *p* < 0.004, *n* = 186, for 1996–2020; r = −0.30, *p* < 0.002, *n* = 132, for 2016–2020). Similarly, hare numbers significantly decreased, but only during the 1996–2020 period (r = −0.47, *p* < 0.0001, *n* = 152).

The spatial distribution patterns of lagomorphs changed statistically after the myxomatosis outbreak (Figure 6 b,c). First, before the start of the epidemic, the space used by hares and rabbits, although statistically different, overlapped extensively (Figure 6a). After July 18, the distribution of both species became much more allopatric. Hares occupied the positive end of the gradient defined by Factor 1 in PCA 2 and PCA 3, while rabbits did the opposite (Figure 6b,c). Rabbits tended to maintain their average habitat (albeit disappearing from many localities), whereas hares were markedly displaced to the south-east (flat farm areas cultivated with wheat and sunflowers), disappearing from many sites within the typical rabbit areas, the hilly sites of the north-west with meadows and olive groves, with some abandoned. A clear spatial segregation between rabbits and hares could thus be observed after the outbreak, which could be conditioned by the differential in hare deaths in the sympatric spaces, and/or by the displacement of hares to myxomatosis-free areas. This process was still more evident when the dates were included in PCA 3 (Figure 6c).

### 3.6. Niche Assemblage Consequences

The niche assemblage analyses between both lagomorph species prior the myxomatosis outbreak in July 2018 showed that 38.56% of the variance was explained by axis 1 (layer 1 and 2 of the CORINE Land Cover) and 20.91% was explained by axis 2 (mainly altitude, with the contribution of layers 2 and 4 of the CORINE Land Cover). Before July 2018, Schoener’s D was 0.24 (see Table 5). The equivalence test showed a non-significant result (*p* = 0.79), whereas the similarity test was marginally significant (*p* = 0.07). After the outbreak, we found that 37.43% of variance was explained by axis 1 (layer 1, 3, and 4 of the CORINE Land Cover) and 21.29% was explained by axis 2 (mainly altitude, and with contribution of layer 2 of the CORINE Land Cover). Schoener’s D was 0.26. The equivalence test gave a non-significant result with a *p*-value of 0.82, while the similarity test was non-significant with a *p*-value of 0.24. (For detailed graphs, see Appendix A). These results indicate that the niches were not more equivalent than expected by chance in any scenario (prior to and after the myxomatosis outbreak), while they were similar before the outbreak, but lost this similarity after it.

## 4. Discussion

Lagomorphs are a fundamental element in Mediterranean ecosystems. The rabbit is considered a typical keystone species, supporting the large, complex Iberian predator community [2]. Similarly, the Iberian hare, although not in the same proportion as the rabbit, is preyed upon by most Iberian predators [16]. However, this endemic species is a key component of the country’s hunting economy, with almost 900,000 individuals hunted per year in Spain in this century [31], although basic knowledge about its biology, ecology, socioeconomic importance, and current conservation status is still scarce (Alves et al., 2002). In this paper, we showed that both rabbit and Iberian hare from a typical Mediterranean agricultural area in S. Spain (1) followed a parallel inter-annual, annual, and daily temporal dynamic, (2) used similar habitats, displaying a highly overlapping distribution pattern, and using habitats with some topographic and crop differences, and (3) substantially shared the environmental niche. The outbreak of a myxomatosis epidemic affecting both lagomorph species in July 2018 has driven important changes in the abundance, dynamics, distribution, and niche, forcing unobserved habitat differences between the rabbit and Iberian hare.

Interactions, especially competitive, between rabbits and hares have received significant attention, but their population consequences tend to be contradictory [10]. High rabbit densities can impact hare populations in several ways, including direct competition for food, shared diseases, attraction of increased numbers of predators to the area, and habitat modification through intense grazing [32]. In some cases, the local disappearance of rabbit populations due to myxomatosis were followed by increases in hare populations (e.g., Britain) or the lack thereof (e.g., Hungary). Data on diet overlap between the two species are also inconsistent, sometimes pointing out or sometimes excluding significant competition for food between them [32]. However, as Flux [10] noted, the rabbit and Iberian hare have coexisted for millennia and they can share the same habitat without major aggressions or strong competition for food because the two species operate at different scales, mediated by the structure of the vegetation, resulting in spatial partitioning or distinct microhabitat preferences [10,33].

### 4.1. Long-Term Population Tendencies of Hare and Rabbit

The rabbit and Iberian hare populations in southern Spain, near Seville, displayed a highly variable dynamic between 1999 and 2021, without any defined tendency. The long-term pattern in hare abundance in Doñana was quite different from that observed here [34]. It may be a consequence of the strong differences in habitat and management of both populations. However, a strong correlation between the yearly relative censuses of both species was observed here. A possible explanation relies on a concurrent response to high-level environmental factors, including primary production controlled by climate and weather conditions, as proposed by Martins et al. [35], crop variety or habitat structure, and predation or epidemics. Both species have similar ecological requirements (e.g., diet and habitat) and both are subjected to similar hunting pressures. A clear example is the fact that the wet spring of 2018 in the area was followed by a slight increase in lagomorph occurrences, possibly due to the higher and longer availability of trophic resources (better and more prolonged grass growth). This relationship between the environment and species dynamics is not exclusive to lagomorphs, with there being many examples across the entire tree of life [36,37]. This result supports the view that the rabbit and Iberian hare can coexist without major aggressions, as Flux [10] pointed out. However, the latest sharp decrease in the relative density of both species coincides with the appearance of the myxomatosis outbreak in hares (2018). A reduction in hare density, although after the outbreak occurrence, was also registered in northern Spain [38]. In this case, a depletion of 62.7% in hare density in two consecutive seasons (2019–2020 and 2020–2021) drove the species to near extinction. What is striking in this case is that the subsequent recovery in rabbits seemed not to have been reciprocated in hares. This may be due to the longer coexistence of rabbits with this disease and their capacity for a quick response through the production of antibodies, which can last more than 18 months [39], in addition to the explosive demography of rabbits [40].

The long-term hunting catches of both lagomorph species in Spain between 2005 and 2018 displayed different patterns, with no correlation whatsoever. While the tendency in rabbits was rather stable, especially in the last decade, a consistent decrease was observed across that period in the Iberian hare (r = *p* < 0.05). This reduction in hare catches has been linked to a reduction in the number of hunters and to other factors related to habitat modifications. A key question, however, is why the decline specifically affected the hare and not the highly relative rabbit. Furthermore, did this tendency bear any relationship with the imminent MYXV hare outbreak in 2018? Abade Dos Santos et al. [41] proposed a hypothesis to explain the 2018 outbreak based on the long-term contact of hares with MYXV, or an antigenically similar virus, since at least 1994. This long-standing contact of hares with MYXV may have occurred with strains that circulated in wild rabbit, or unnoticed strains circulating in Iberian hare populations.

### 4.2. The Myxomatosis Outbreak (2018–2020)

The first clinically confirmed cases of myxomatosis infection in Iberian hares were observed on 20 June 2018 in the province of Cuenca (central Spain), and newly affected sites were progressively aggregated from southern and central regions [22]. As the mean time interval between the first and maximum number of hares detected with myxomatosis was 31 days, the start of the epidemic in the Carmona countryside may well be before July 18, as this was the date when we stopped observing hares. The spatial distribution of the initial hare mortality was quite homogeneous throughout the affected areas in the province of Cordoba (S. Spain), according to García Bocanegra et al. [15], where the first detected case was on 10 July. Thus, if we observed the consequences of the mortality from July 2018, this homogeneity may be more spatially generalized than supposed. Could the outbreak have begun simultaneously in two sites 100 km apart? Another interesting question posed by our results stems from the coincidental almost complete disappearance of both rabbit and Iberian hare in our study area, whereas such simultaneity of occurrence of the disease in both species was registered in only 27% of the areas analyzed by García Bocanegra et al. [22]. Was it a mere coincidence or could there be a relationship, e.g., virus transmission, from one species to the other? Our results suggest a slight niche similarity and spatial coincidence between the species prior to the outbreak in 2018, but no spatial coincidence or niche similarity between them after it. Therefore, what kind of interaction may have fostered the transmission between hare and rabbit in southern Spain?

It is well-established that disease is a major source of death in hares [42]. Hare populations are at their highest densities in autumn, with a high proportion of juveniles [42]. These authors describe that at that time of year, disease leads to heavy mortality, suggesting that concentration of hares, especially juveniles, into remaining foraging areas in the autumn may promote intraspecific disease transmission. Similarly, in northern Spain, the highest mortality among Iberian hares due to myxomatosis was in October [38] The displacement ability of hares between different habitats is well-known [43,44]. This capacity allows this species to recolonize wide unpopulated areas in drought years. The most important transmission route of the most studied disease in hares, the European brown hare syndrome (EBHS), is oral-fecal, although oral and nasal transmission also occurs [45]. These authors suggest that the main route of infection of myxomatosis in hares is the virus transmission through blood-feeding insects or close contact between hares and infected rabbits and their excretions. In rabbits, the main route of transmission is through biting arthropods (fleas) or by direct contact [46].

Although it is clear that a jump of MYXV occurred between rabbit and hare , and that the virus has the capacity for this interspecific transmission, exactly how the jump of myxomatosis from rabbits to hares occurred is unknown, and thus the dynamics and interaction between the two species are essential to identify the cause. In our case, was there any connection between the more or less simultaneous disappearance of rabbits and hares? Was there any possibility of direct transmission from one species (rabbit) to another (hare)? Why did the observed spatial segregation of the two lagomorphs occur after the myxomatosis outbreak? In the absence of relationships between rabbit and hare myxomatosis, the outbreak in hares should have been independent from rabbits. Additionally, it should not have affected the spatial distribution of both species. One aspect to take into account is the trophic behavior of these lagomorphs.

Our results support the hypothesis of a direct transmission from rabbits to hares via fleas, blood-feeding insects, or a similar mechanism. First, there was a clear spatial segregation of both species before and after the outbreak. Second, after the niche analysis, we observed a higher niche similarity before the outbreak, but a lower niche similarity after it. According to the data presented here, the myxomatosis could have originated first in rabbits, and was then transmitted via fleas, mosquitoes, or some other alternative, to hares coexisting with ill rabbits. Both the infected hares and rabbits died or disappeared. As a consequence, the only remaining lagomorphs in the study area were rabbits with an adequate immune response to myxoma virus and hares inhabiting sites with no contact with rabbits.

### 4.3. Annual Cycle

It was described that hares feed at night, gathering in small groups and sometimes coinciding in time and space with rabbits [7]. Regarding the breeding period of both species, hares always start breeding in the same month across all countries (January with the peak of newborns in March–April, as described by Alves et al. [47], or the equivalent July in the Southern Hemisphere). This suggests that the main driving factor controlling the breeding period is the photoperiod [48] and not any other environmental factor such as temperature [49]. In rabbits, on the other hand, it is known that the breeding season is adaptable, especially in Australia, where it depends on the irregular rainfall, therefore suggesting a dependence on the availability of green grass [50,51]. After the myxomatosis outbreak in hares (July 2018), we observed a clear disruption of the annual pattern in the case of this species, contrasting with the case of rabbits where the annual cycle seemed not to have been affected.

### 4.4. Daily Cycle

Rabbits concentrate their daily activity into the twilight, the night, and the sunrise, thus varying the specific hours between seasons [52,53]. The pattern is similar in the case of the European hare (*Lepus europaeus*) [54], and exclusively nocturnal in the Iberian hare studied in Doñana National Park [55] and Villafáfila [56]. In rabbits, the described factors mediating the activity rhythms are temperature, sunlight, moonlight, and wind, with maximums of daily activity in December [57,58]. In the case of hares, the activity concentrates into the two hours prior to sunrise, and ceases just before it [56]. Our results showed agreement with this information, with a similar pattern between species, although with approximately an hour of difference in the cessation of activity throughout the year (earlier in Iberian hares). This matches the described strictly nocturnal behavior of Iberian hare.

Our results showed that species variability and date had a clear influence on the daily cycle during the period from June 2017 to January 2018, whereas the relation was less clear from March 2018 to July 2018 (see Table 2). This could be explained by the low detection rate, and hence low statistical power, following the myxomatosis derived mortality.

### 4.5. Spatio-Temporal Relationships

As previously mentioned, lagomorphs are a basic element in the trophic chain of the Iberian Peninsula [15], being the basic prey of many species [3,4,5,6]. Thus, understanding the dynamics related to their decline is of great interest for the conservation of Iberian wildlife. Previously widespread in rabbit populations [18], the recent appearance of myxomatosis in Iberian hares has raised concern not only in the conservationist world but also in other sectors such as sport hunting [21]. It seems clear that the conditions necessary for the intraspecific transmission are not common since the disease has been present in rabbits for decades without any evidence in hares. Laguna et al. [59] proposed the spatially explicit aggregation index (SAI) as a measure of aggregation between individuals and between groups, assuming that higher values, and thus higher aggregation, would ease transmission. However, this index was applied to ungulates, and our results showed a spatial segregation between individuals of both species in agricultural land, as noted by other authors in other habitats, and for other but phylogenetically related hare species [11]. This fact makes transmission by direct contact difficult, although it is known that the main infection route of the virus is blood-feeding arthropods and contact with infected excrement [45]. Our data also showed certain overlap, as the rabbit is more generalist than the hare. This seems to reinforce the above-mentioned main infection route through blood-feeding insects and feces, especially considering the double digestion of lagomorphs (the first, soft feces are re-ingested for a second digestion).

García-Bocanegra et al. [22] observed that the spatial distribution of ha-MYXV infection cases within the affected areas was homogeneous in most of the areas surveyed. In our case, there was a strong spatial effect of myxomatosis, such that hares disappeared from sites where this species coexisted with rabbits. Does it mean that the latter were responsible for the transmission of the virus to hares?

### 4.6. Niche Assemblages

Regardless of the strict spatio-temporal relationship between the two species, we analyzed the niche occupied by both, based on land cover and altitude variables. The most interesting result was that although the niche overlap was always around 0.25 (Schoener’s D), prior to the myxomatosis outbreak, niches were similar between both species without being equivalent. This clearly matches with the results of Scott [60], who described a similar habitat, although with differences at a fine scale. However, after the 2018 outbreak, the remaining pockets of hare in Mediterranean agriculture arations occupied only a niche which was dissimilar to that of rabbits. This clearly suggests that hares have persisted in these areas because there was no contact with infected rabbit excrements, and the exchange of blood-feeding insects between the species was more difficult, especially considering the local scale of the study area. Additionally, similar patterns regarding the trophic niche were observed by Lush et al. [61] with respect to our results prior to the outbreak. In their study of rabbits and brown hares (*Lepus europaeus*), the authors found similarities in the main grasses consumed by both, but with a clear partitioning of the trophic niche with respect to the rest of the dietary components, which was possibly due to the habitat segregation. This slight coincidence in niche between both genera seemed to disappear in our study area after the outbreak, thus suggesting a negative impact of rabbits on hares when myxomatosis is circulating.

## 5. Conclusions

The main conclusion we could draw was the exclusion after the outbreak of hares in areas populated with rabbits, possibly because the surviving hare populations remained only in areas without rabbits, and thus avoided contagion through blood-feeding insects or contact with excrement.

As a key element in Mediterranean ecosystems, the collapse—albeit temporary—of rabbit and hare populations in agricultural land may have important consequences for the trophic web with severe conservation derivatives. We proposed a hypothesis for the main effects of the observed lagomorph crash on selected elements from the web of this highly anthropized site. Summarizing our hypotheses for these effects, we propose the scheme shown in Figure 7. After the outbreak, the depletion of the lagomorph population may cause a decrease in the number of raptors and carnivore mammals [62,63]. This decrease in predator numbers, and the consequent increase in the primary production caused by the lack of foraging lagomorphs, should generate an increase in other groups such as galliforms (for instance, the red partridge, *Alectoris rufa*) and other prey species [64]. Other groups depending on primary production, such as Lepidoptera, may also increase. These ecological consequences appear similar to those observed in other latitudes, such as those described by Sumption & Flowerdew [65] in the British Isles. Although our hypotheses have yet to be tested, we can expect a similar pattern.

Finally, mention should be made of the socioeconomic impact of the depletion in the Iberian hare population, as it is traditionally considered a noble hunting piece that supports an important economic chain through hunting societies, trained dogs, game wardens, competitions, etc. Here, a new hypothesis arises: it is well-known that myxomatosis was introduced into Europe in 1952 [66], and has circulated in rabbits since then. Taking into consideration the 70-year coexistence with the disease, the higher densities of both rabbits and hares in the past, and the weak overlap between the two species, we hypothesize that the recent outbreak in hares might be related to a favored contact mediated by human game management, as in breeding farms and transportation during introductions. Of course, this is an untested hypothesis that needs further research. Additional research is also needed with respect to the synergic factors contributing to the Iberian hare rarefication, for example, regarding the effects of agrochemicals. Martínez-Haro et al. [67] found that up to 22% of hares killed in hunts, and up to 45% found dead, had values of up to 16 μg/g wet weight of glyphosate. Understanding the dynamics of both species, rabbits and hares, after the disease outbreak is thus essential, not only for biology conservation purposes but also for economic and traditional aspects. Collaboration between scientists, conservationists, and hunter societies can produce a positive synergic impact on the interests of all sectors involved.

## Figures and Tables

**Figure 1 animals-12-01351-f001:**
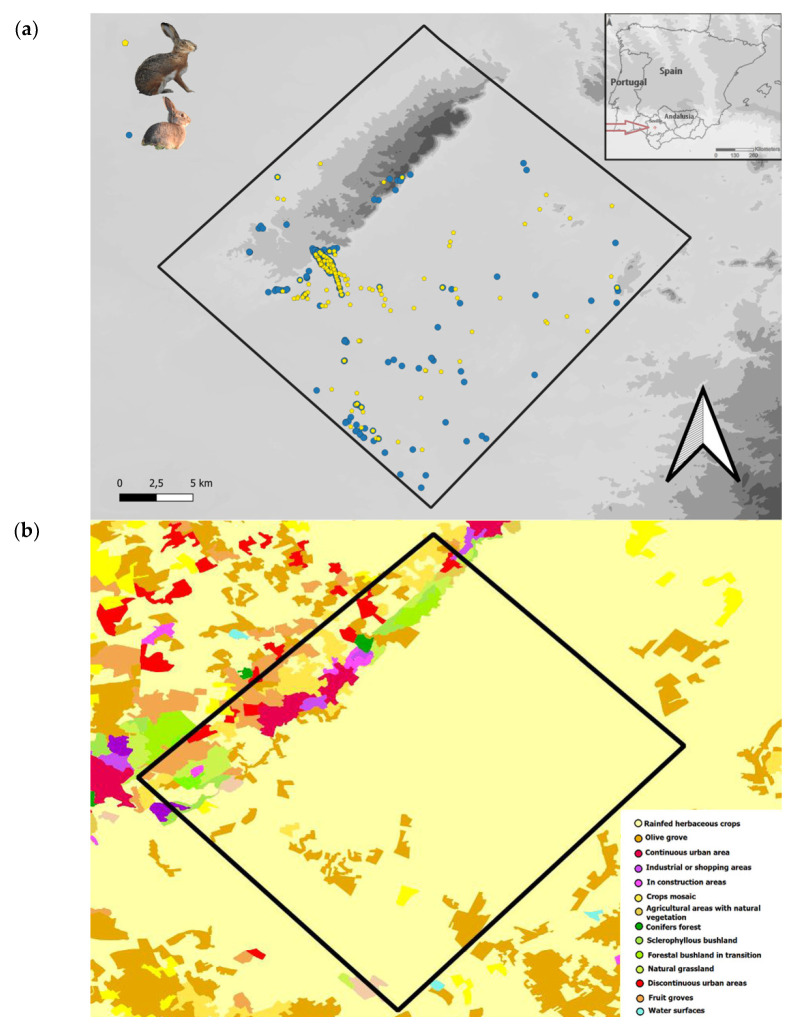
Map of “Vega de Carmona”, “Los Alcores”, and surrounding area in the province of Seville (S. Spain). The box marks the study area. (**a**) Dots represent the recorded localities of Iberian hares (yellow) and rabbits (blue) over the digital elevation model layer. (**b**) Land uses of the study area.

**Figure 2 animals-12-01351-f002:**
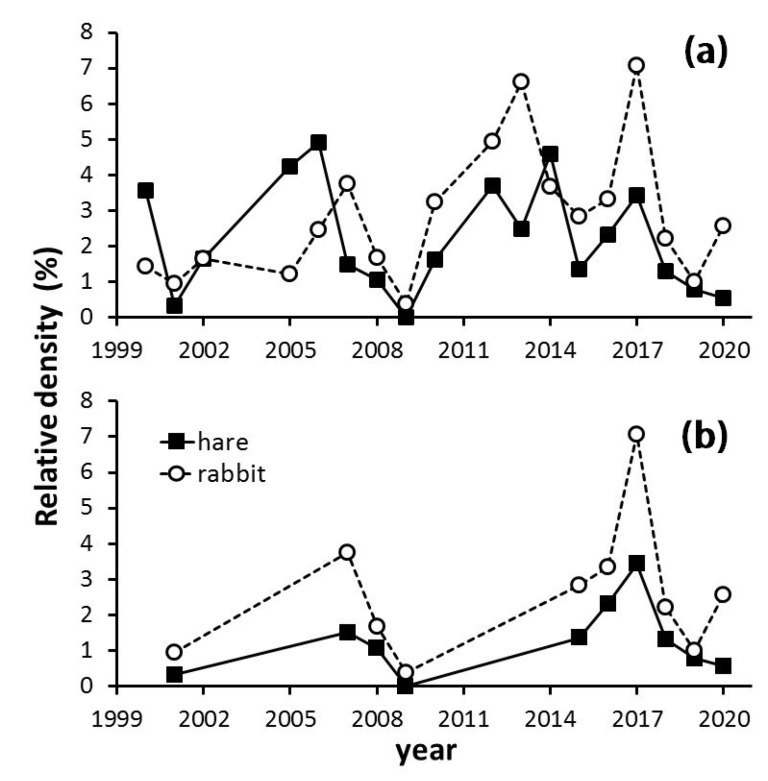
Temporal dynamics of the Iberian hare (*Lepus granatensis*) and rabbit (*Oryctolagus cuniculus*) in a Mediterranean agricultural area with high human population (S. Spain). Relative density is calculated as the number of individuals registered per year, corrected by the total number of localities × date × year in which registers were made (×100). (**a**) Years with more than 120 localities × date (*n* = 18). (**b**) Years with more than 300 localities per date (*n* = 10).

**Figure 3 animals-12-01351-f003:**
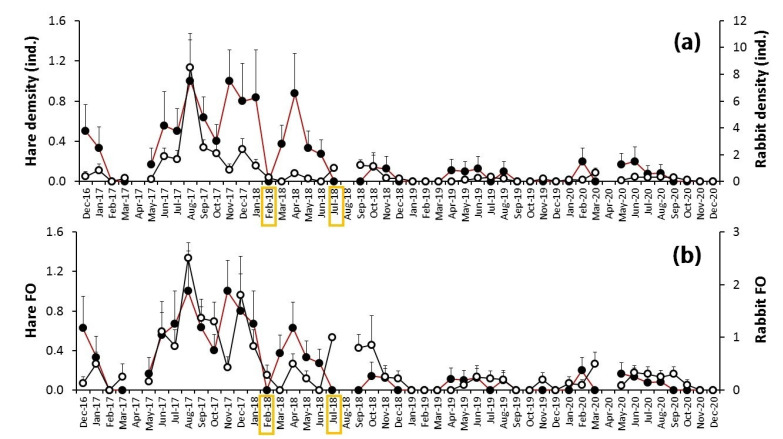
Evolution of (**a**) the monthly average relative density (individuals) (±SE) and (**b**) the average frequency of occurrence (±SE), based on the registers of the Iberian hare (*Lepus granatensis*) (black dots) and rabbit (*Oryctolagus cuniculus*) (white dots) in a Mediterranean agricultural area (S. Spain) between December 2016 and December 2020. Yellow squares mark the important events.

**Figure 4 animals-12-01351-f004:**
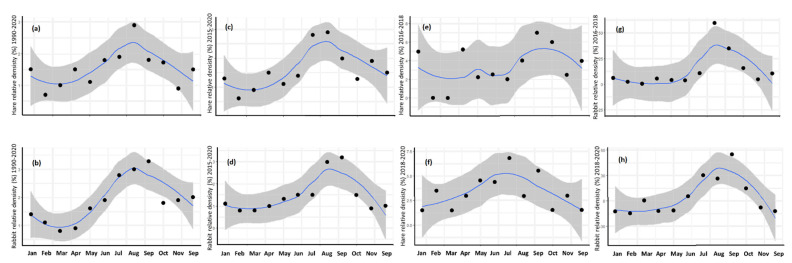
Monthly evolution of the relative density (%) of the hare (*Lepus granatensis*) and rabbit (*Oryctolagus cuniculus*) registered in a Mediterranean agricultural area (S. Spain). (**a**,**e**) Data from the full study period (1995–2020); (**b**,**f**) data from the period 2015–2020, in which a more intensive sampling was carried out. (**c**,**g**) Monthly dynamics for the hare, considering two intensively sampled periods: December 2016–June 2018, prior to the start of the epidemic, and July 2018–December 2020, after the myxomatosis outbreak. (**d**,**h**) The same for rabbit. Statistic: smooth.

**Figure 5 animals-12-01351-f005:**
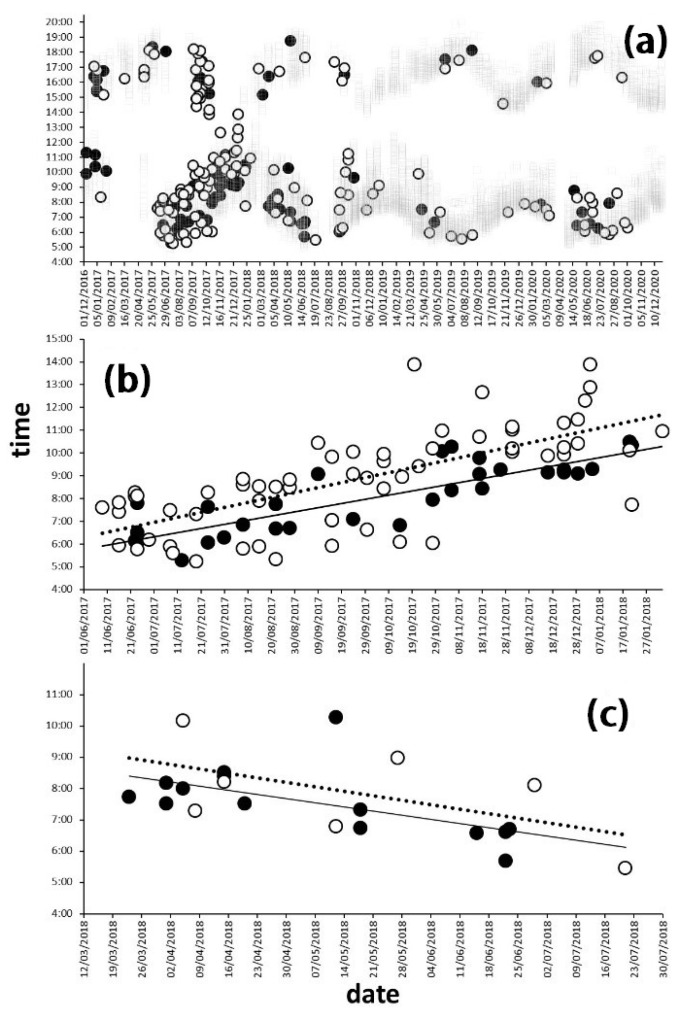
Observation times of the Iberian hare (*Lepus granatesis*) (black dots) and rabbit (*Oryctolagus cuniculus*) (white dots) during the period in which quantitative surveys were taken between December 2016 and December 2020. (**a**) The grey squares represent every sampling point. (**b**) Only morning surveys between June 2017 and February 2018. (**c**) Surveys taken before noon, between March and July 2018. In (**b**,**c**), the adjustments between date and time are represented by a solid line for hares and a dotted line for rabbits.

**Figure 6 animals-12-01351-f006:**
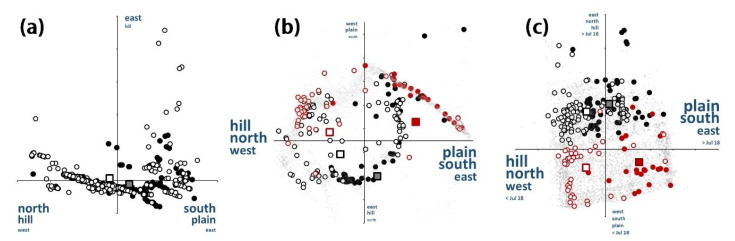
Distribution in the space defined by the two main factors of a principal component analysis (PCA) applied to a matrix where the rows are sites and the columns are the X, Y, and Z (altitude, m) coordinates of each record, plus date, in the case of (**c**). (**a**) PCA 1: every dot represents a location for the Iberian hare (*Lepus granatensis*) (black dots) and/or rabbit (*Oryctolagus cuniculus*) (white dots) registered between 1995 and 2019 (*n* = 535), (**b**) PCA 2. (**c**) PCA 3. In addition to the axis, the interpretation of each factor is indicated. The size of the words tends to be roughly proportional to the importance of each feature (see Table 3 for details). The squares represent the centroids of each species or species and date distribution. In (**b**,**c**), solid dots are hare records (black indicating those made prior to July 2018, when the myxomatosis outbreak started in the study area, and red for those after July 2018). Empty dots represent rabbit records (black prior to July 2018 and red after July 2018). Grey dots: points without lagomorph records. *n* = 5867 total points (88 hares and 132 rabbits).

**Figure 7 animals-12-01351-f007:**
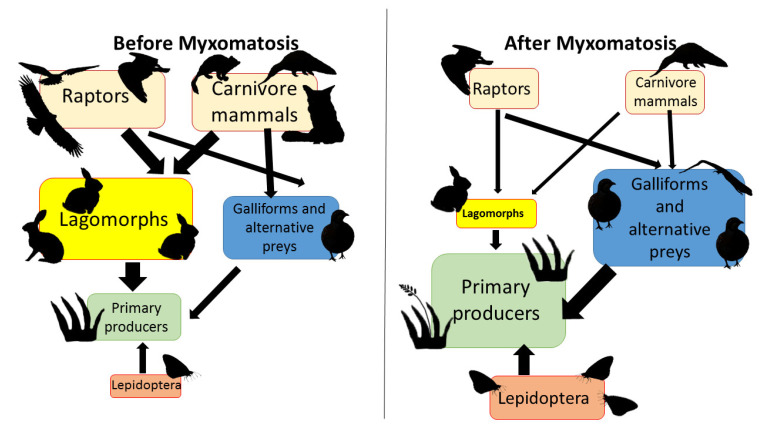
Scheme of the impact of lagomorph depletion in the trophic chain of a pseudo-steppe agricultural land. The direction of arrows indicate which group predates or consumes over the other.

**Table 1 animals-12-01351-t001:** Spearman’s rank-order correlations (in bold) between the temporal dynamics (December 2016 to December 2020), measured as mean monthly relative density and mean monthly frequency of occurrence (FO), of the Iberian hare (*Lepus granatensis*) and rabbit (*Oryctolagus cuniculus*) in a Mediterranean agricultural area in S. Spain (*n* = 46). Intraspecific correlations between density and FO are included only to confirm both as alternative measures.

	Spearman r_s_	t (N-2)	*p*-Level
Hare density–hare FO	**0.98**	30.77	0.000000
Rabbit density–rabbit FO	**1.00**	73.03	0.000000
Hare density–rabbit density	**0.54**	4.29	0.000096
Hare FO–rabbit FO	**0.50**	3.80	0.000440
Hare density–rabbit FO	**0.50**	3.84	0.000387
Hare FO–rabbit density	**0.54**	4.30	0.000095

**Table 2 animals-12-01351-t002:** Observation times of the Iberian hare (*Lepus granatensis*) and rabbit (*Oryctolagus cuniculus*) registered in a Mediterranean agriculture area in S. Spain. The data correspond to the two periods represented in Figure 6b,c. Signif. codes for Glm: 0 ‘***’, 0.001 ‘**’, 0.01 ‘.’, 0.1 ‘ ’, and 1.

		17 June–18 January				18 March–18 July			
		Hare		Rabbit		Hare		Rabbit	
Median		8:08:56		8:49:23		7:31:01		8:06:43	
Minimum		5:16:50		5:14:31		5:41:36		5:27:53	
Maximum		10:29:14		13:53:10		10:16:53		10:10:02	
Cv (%)		18.7		24.5		14.9		19.4	
*n*		32		61		14		7	
Glm (time~date + species)		Estimate	Std. Error	T value	Pr(>|t|)	Estimate	Std. Error	T value	Pr(>|t|)
	Intercept	−3.652 × 10	3.694	−9.886	4.92 × 10^−16^ ***	3.554	1.909	1.861	0.0642 .
	Date	8.560 × 10^−4^	8.586 × 10^−5^	9.970	9.970 3.30 × 10^−16^ ***	−7.209 × 10^−5^	4.438 × 10^−5^	−1.625	0.1058
	Species	4.024 × 10^−2^	1.246 × 10^−2^	3.229	0.00174 **	−2.621 × 10^−3^	2.430 × 10^−2^	−0.108	0.9142

**Table 3 animals-12-01351-t003:** Eigenvalues of correlation matrices and percentage of total variance explained by the first two factors of the three principal component analysis applied to the distribution data of hares and rabbits from a Mediterranean agricultural area in S. Spain.

PCAs	Factor	Eigenvalue	% Total Variance	Cumulative Variance (%)
(1) Coordinates (x, y, z) of lagomorph observations during the entire study period (1995–2020)	1	1.50	50.1	50.1
	2	1.02	33.9	84.0
(2) Coordinates (x, y, z) of every sampling point during the intensive survey period (2016–2020)	1	2.28	76.1	76.1
	2	0.62	20.7	96.8
(3) Coordinates (x, y, z) plus date of every sampling point during the intensive survey period (2016–2020)	1	2.34	57.8	57.8
	2	1.03	25.9	83.7

**Table 4 animals-12-01351-t004:** Pearson correlations between the three main factors of a three principal component analysis (PCA) applied to a matrix of locations × UTM coordinates, altitude, and eventually, date, where Iberian hare (*Lepus granatensis*) and rabbit (*Oryctolagus cuniculus*) could be registered from a Mediterranean agricultural area in S. Spain. PCA 1: the rows are records of Iberian hare and rabbit made between 1995 and 2019, while the columns give the X and Y coordinates, and the altitude (m) of each record. PCA 2: locations × UTM coordinates and altitude visited between December 2016 and March 2020. PCA 3: the same as previously, but including the date as a new variable. All the variables were log-transformed. The date was also correlated with Factors 1 and 2 from PCA 1 and 2 to check whether theses gradients were also temporal. * *p* < 0.001.

	PCA 1 (*n* = 535)		PCA 2 (*n* = 5867)		PCA 3 (*n* = 5867)	
	FACTOR 1 (50.1%)	FACTOR 2 (33.9%)	FACTOR 1 (76.1%)	FACTOR 2 (20.7%)	FACTOR 1 (57.8%)	FACTOR 2 (25.9%)
UTMx coordinate	0.37 *	0.90 *	0.74 *	−0.24 *	0.73 *	0.68 *
UTMy coordinate	−0.87 *	−0.02	−0.91 *	−0.19 *	−0.91 *	0.35 *
Altitude (m)	−0.77 *	0.46 *	−0.94 *	−0.23 *	−0.96 *	0.19 *
Date	−0.36 *	−0.01	0.22 *	−0.94 *	0.13 *	0.19 *

**Table 5 animals-12-01351-t005:** Schoener’s D and *p*-values of equivalence and similarity test under conservatism hypothesis. Contribution of variables also provided. Bio1–Bio4: four levels of CORINE Land Cover.

	Variable Contribution	Schoener’s D	Equivalence Test	Similarity Test
Prior to the myxomatosis outbreak	Bio1 (0.24); Bio3 (0.27); Bio2 (0.13); Bio4 (0.18); altitude (0.18)	0.24	*p* = 0.79	*p* = 0.07
After Myxomatosis outbreak	Bio1 (0.27); Bio3 (0.19); Bio2 (0.17); Bio4 (0.14); altitude (0.23)	0.26	*p* = 0.82	*p* = 0.24

## Data Availability

The data are available upon motivated request to the authors.

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
