# Peer review of "The Dynamics of Lepus granatensis and Oryctolagus cuniculus in a Mediterranean Agrarian Area: Are Hares Segregating from Rabbit Habitats after Disease Impact?"

_animals, 2022, doi:10.3390/ani12111351_

Round 1
Reviewer 1 Report
Manuscript ID: Animals-1724415
Title: Dynamics of Lepus granatensis and Oryctolagus cuniculus in a Mediterranean agrarian area: Are hares segregating from rabbit’s habitat after disease impact?
The manuscript " Dynamics of Lepus granatensis and Oryctolagus cuniculus in a Mediterranean agrarian area: Are hares segregating from rabbit’s habitat after disease impact?" shows an interesting study. It should note the difficulty of this study, since the data were not obtained from an experiment under controlled environment and authors required a long time to collect a representative data base. Prior to recommendation of the manuscript for publication, major revision is necessary.
Abstract:
Please, add more information about used methodology and analyzed traits. The abstract ends with a short and clear conclusion from your data.
Keywords:
Ok
Introduction:
OK
Methods:
In this section, authors should avoid the use of quotation marks and question marks.
Additional information on the methodology used in data collection should be included. For example, how the indirect signs (tracks, faeces or burrows) were differentiated whether they were from hares or rabbits? were the number of pieces hunted and the cases of myxomatosis and VHD in the population of hares and rabbits count? etc.
Authors indicate that they used a generalized linear model (Gaussian) but they didnot give information about which variables were considered as Y and X and about fixed factors.
In the legend of Figure 1, please replace red for yellow.
Results:
Authors must remember that in this section the results obtained from the analysis of their data must be shown, not from other works such of García-Bocanegra et al., 2019).
Figure 2, why do authors arbitrarily take the values for years > 120 and > 300?. This point needs to be clarified in Methods.
It should be noted that authors have not registered the cases of myxomatosis in their population, so it is very difficult to state categorically that the drop in the animal census in 2019 is due to myxomatosis. It could also be due to VHD, or hunting, or the farming system, etc. Anyway, this is a point to move to discussion.
Authors comment an important reduction in rabbit and hare population on 2019, but there was other in 2009. What happened in 2009? Why it was not commented?.
Discussion:
Please, delete Figure 7.
Do not ask questions in the document, use indirect speech.
Conclusions:
The conclusions section must be rewritten because it is very long.
Note that the conclusions must came from this study and not from other works. Therefore, all citations must be removed from it.
Author Response
REVIEWER 1
The manuscript " Dynamics of Lepus granatensis and Oryctolagus cuniculus in a Mediterranean agrarian area: Are hares segregating from rabbit’s habitat after disease impact?" shows an interesting study. It should note the difficulty of this study, since the data were not obtained from an experiment under controlled environment and authors required a long time to collect a representative data base. Prior to recommendation of the manuscript for publication, major revision is necessary.
Thank you very much for your useful comments and recommendation. We have addressed all your comments below:
- Abstract:
Please, add more information about used methodology and analyzed traits. The abstract ends with a short and clear conclusion from your data.
Expanded as proposed:
Lagomorphs, with two genera inside the Iberian Peninsula, Oryctolagus and Lepus, are an essential element of the trophic chain in this region, being the main prey of many predators, including some highly endangered such as the Iberian lynx (Lynx pardinus). Myxomatosis, a disease produced by a myxoma virus producing tumours in conjunctive tissues, has caused mass mortalities in rabbits (Oryctolagus cuniculus) for decades. Recently, the virus has jumped interspecifically from rabbits to Iberian hares (Lepus granatensis), and this has originated a severe depletion of hare populations, generating great social and ecological concern. We have studied the population dynamics and distribution of both lagomorph’s species in a Mediterranean agricultural area from southern Spain with a combination of non-systematic and systematic field data collected since 1990s. The appearance of the myxomatosis outbreak in the Iberian hare in 2018 allowed us the opportunistic analysis of the spatial and numerical consequences of this new epizootic on rabbits and hares, based on univariate and PCA analyses. The effects of this sudden lagomorph’s mortality on their respective daily and seasonal cycles were also studied, as relations with the temporal dimension using generalized lineal models (GLM). In the 2018 outbreak, we first observed a drastic reduction in rabbits and then in hare registers. But this reduction in lagomorph abundance had the largest population and spatial consequences on Iberian hare, but not on rabbit. Before the mortality event we observed some spatial overlap and niche similarities between both lagomorphs, but after June 2018 hares almost disappeared from the sites initially shared with rabbits. Both, the outbreak time schedule and its specific spatial consequences, suggest a direct transmission of myxoma virus from rabbits to hares, due to a close contact of both species. The different population responses observed in rabbits vs. hares, i.e. the population recovery of rabbits after several months of the outbreak, is pointing out to the lack of contact of hares with myxoma virus and the absence of an adequate immune response of this species. Finally, we provide an overview of this hare population depletion on the ecological and socio-economic dimension of the Iberian Mediterranean region, where lagomorphs, also hares are keystone species.
- Keywords:
Ok
- Introduction:
OK
- Methods:
In this section, authors should avoid the use of quotation marks and question marks.
Done
- Additional information on the methodology used in data collection should be included. For example, how the indirect signs (tracks, faeces or burrows) were differentiated whether they were from hares or rabbits? were the number of pieces hunted and the cases of myxomatosis and VHD in the population of hares and rabbits count? etc.
The identification of indirect signs was based in an extensive field experience of the authors and in classic guides for Iberian mammals, as those from Benjamin Sanz. We have included this information in the paper, as follows:
…the presence/abundance of Lagomorpha, based on direct observations or indirect signs (footprints, faeces or burrows), identified by the authors, following an extensive field experience and criteria as those from Sanz (1997)….
Myxomatosis data were provided by local hunters, confirmed by governmental officers. Hunters from Arahal, a town within the study area, took field samples (carcasses or ill animals) that were sent to the veterinary services of the Junta de Andalucía, that confirmed the myxomatosis. Later, these technicians also visited the area and took new samples of hares affected by myxomatosis.
Later in the paper, we explain this about the myxomatosis data.
- Authors indicate that they used a generalized linear model (Gaussian) but they didnot give information about which variables were considered as Y and X and about fixed factors.
We have reworded it as follows:
Additionally, in order to test the effects of date and species in the daily activity time, we used a generalized lineal model (Gaussian), being time (daily hour) the dependent variable, and date and species (fixed factor) the variables.
- In the legend of Figure 1, please replace red for yellow.
Done
Results:
- Authors must remember that in this section the results obtained from the analysis of their data must be shown, not from other works such of García-Bocanegra et al., 2019).
We agree with the reviewer view, but in this specific case the only published information on the myxomatosis outbreak is that of García-Bocanegra et al. (2019). Anyway, we have added the information obtained from the local game club, from Arahal (Sociedad Deportiva de Caza Arahalense) on the reason and date of the outbreak.
The question about the date of the outbreak and about the coincidence of the decline between rabbit and hare in the study area, have been moved to the Discussion.
- Figure 2, why do authors arbitrarily take the values for years > 120 and > 300?. This point needs to be clarified in Methods.
We have explained in methods that the reason to arbitrarily select years with N>120 or 300 was to avoid spurious relationships due to low sample size. It has been reworded as follows:
To avoid the effect of spurious relationships due to low sample size, we have arbitrarily considered only years with N> 120 localities x date or, ideally, with N>300 localities x date.
- It should be noted that authors have not registered the cases of myxomatosis in their population, so it is very difficult to state categorically that the drop in the animal census in 2019 is due to myxomatosis. It could also be due to VHD, or hunting, or the farming system, etc. Anyway, this is a point to move to discussion.
As replied in question 8, we have reworded this point in the results and moved to the discussion the reflections on dates and coincidence of the outbreak on both species.
We have maintained the reference of García-Bocanegra et al. (2019), because there were hares analysed in the study area for local game clubs in the same period positive for myxomatosis and as such were considered by the Government. And also because this reference, the only one published available, matches in time and it is very close in distance with our data.
Additionally, the mortality was parallel in both species and VHD is not registered as causing mortalities in Lepus granatensis.
- Authors comment an important reduction in rabbit and hare population on 2019, but there was other in 2009. What happened in 2009? Why it was not commented?
This is a really interesting point. The close correlation in the dynamics between both species is stunning. And the two minima before the one during the outbreak in 2018 remains a mystery for us. We have reviewed the original field sheets. The sites visited throughout the study period were not always the same, but in 2009 were in areas especially rich in Lagomorphs. Climatically, 2009 was an average year for the study area (541 mm rainfall, 20 average temperature). In this year, it was reached the maximum number of hares hunted in Spain (see Fig. 7). We don’t have any specific environmental evidence on the causes that may have produced such a minimum, excepting, perhaps, a high hunting pressure in the area that may have produced this numerical reduction. But this has to be also speculative because there are no data available on the number of hares and rabbits locally hunted.
Fábio A. Abade dos Santos, in a preprint from 2020, show from serological analysis that L. granatensis have been in contact with myxoma virus for decades. While preparing this manuscript we have speculated on this possibility. Why not a previous contagion of hares from rabbits with myxoma virus in the area? A low intensity illness in hares, but enough as to rarefy this species, to produce a minimum in 2009. But this is even more speculative than the previous environmental hypothesis.
For this reasons –a lack of solid evidences on potential factors explaining the observed dynamics- we have not commented on this result.
Discussion:
- Please, delete Figure 7.
Deleted.
Conclusions:
- The conclusions section must be rewritten because it is very long.
Note that the conclusions must came from this study and not from other works. Therefore, all citations must be removed from it.
We agree, this section is really a general discussion of the implications of the manuscript. We have changed it to discussion section that better fits with their content.

Reviewer 2 Report
The work is interesting, although too extensive for the question addressed. Authors ask too many questions throughout an article and this is not usual or expected in a scientific article. English needs a major overhaul and there are some references missing.
Line 11 . Lagomorpha includes mucho more than two genera. I understand that you are talking about Peninsula Iberica but this is not clear in this sentence. In other way the sentence appearseems to say that these genera only exist in the Iberian peninsula. Should appear : “The genera Oryctolagus and Lepus (order Lagomorpha) are an essential element of the trophic chain in Iberian Peninsula, being the main prey….”
Line 37 “L. granatensis is the widest distributed hare species in the Peninsula, and the only one present in the south of Spain (Palacios & Mejide 1979) and in Portugal”
Lines 62 to 65 – This paragraph is not well written and is not scientific-like language. What is “veterinary criteria”?
Line 73 : herbicide is a pesticide. Remove the first.
Line 77 : rabbit hemorrhagic disease (RHD)
Line 79 : The authors should use official information and criteria, not subjective criteria. The lynx is endangered (IUCN) and not “highly endangered” and the Iberian imperial eagle is Vulnerable (IUCN).
Figure 1 – The hare points are yellow, not red
Line 178 – Final point missing
Line 214 – “Could this reduction be attributed to the same infectious 214 outbreak observed in Córdoba?” Questions should not appear in the results. Remove.
Line 373 – “outbreak of a myxomatosis epidemics affecting to both lagomorph species in July 2018”. The same epidemic outbreak is not treated, keeping in mind that the viruses form differently for the species. The virus ha-MYXV (or MYX-Tol) was only described later in coelho-bravo (Abade dos Santos, 2020). Reformulate this sentence.
Line 380 “In some cases, the 380 local disappearance of rabbit populations due to myxomatosis were followed by increases 381 in hare populations (e.g. Britain) or not (e.g. Hungary). ”. Include the reference or remove.
Line 411 – The authors should not talk about myxomatosis caused by classic MYXV and ha-MYXV like they knew that the diseases are the same and have the same characteristics. In fact, viruses behave completely differently (Evaluation of commercial vaccines (https://doi.org/10.3390/vaccines10030356), and it was known not long ago (Abade dos Santos, 2022. Evaluation of commercial vaccines…) that antibodies produced against the classic virus protect against the new ha-MYXV virus, which is naturally recombinant, but it is not certain that they protect during 18 months. Rephrase this paragraph accordingly.
Line 422 – This author is usually referred as “Abade dos Santos”. Please confirm
Figure 7 – Species name should appear italicized
Line 452 – “It is known that disease is a major source of deaths in hares (Edwards et al. 2000)” Which disease?
Line 472 – The same article referred before (Evaluation of commercial vaccines) proved that rabbits are highly susceptible to both natural recombinant virus (ha-MYXV) isolated from wild rabbit and from Iberian hare, so it is clear that transmission could appear between these two species. In fact, the transmission does not need to be direct, being much more likely an indirect transmission (insects, fleas...)
Line 477 to line 480 – Remove the paragraph. It is well known that hares do not eat meat.
Line 481 to 483 – I can not understand what authors are hypothesizing. These species do cecotrophy but it not means that hare eats rabbit faeces. The process is the ingestion of the cecotrophs directly from the anus of the animal itself, not from the ground.
Line 490 – “This is com- 490 patible with the hypothesis that the remaining hares after the mortality are those avoiding 491 the contact with areas used by rabbits.”
This sentence is totally speculative. In fact, what happens is that animals do not like to compete for food and space, and hares are territorial animals. However, when there are high densities of both species it is normal for them to share the habitat, when there are few specimens it is normal for both species to retract to places with more food density, that is, not inhabited by other species.
Line 509 – Lepus europaeus
Figure 7 – It is not clear if authors are trying to represent a trophic chain. In any case, the direction of arrows is confused.
Author Response
REVIEWER 2
- The work is interesting, although too extensive for the question addressed. Authors ask too many questions throughout an article and this is not usual or expected in a scientific article. English needs a major overhaul and there are some references missing.
This is a good point about our work that needs to be explained. Thanks for asking it. This paper is about the way some rabbit and hare populations behave temporally and spatially in a Mediterranean agricultural land highly populated by humans. Up to we know, this question has not been directly addressed. There are no papers on it. Overlapped on this study we unpredictably met with a myxomatosis outbreak affecting both lagomorph species. This completely original circumstance provides a unique opportunity to describe the impact of the outbreak on the temporal and spatial dynamics of hare and rabbit. And at the same time it allows us to delve into the way hare and rabbit interact and, most important, about how the hare outbreak could be produced and on the potential mechanisms involved in the myxoma virus transmission from rabbits to hare.
The necessary combination of a sound description of the dynamics of both species, altogether with the knowledge of the impact of myxomatosis and its consequences on how hares and rabbits interact makes it essential to maintain the work in its current form. The cutting of any part of the work will leave unaccomplished the main goals of the paper. One exception may be Fig. 7 as it can be considered only partially informative, thus we have eliminated it.
- Line 11 . Lagomorpha includes mucho more than two genera. I understand that you are talking about Peninsula Iberica but this is not clear in this sentence. In other way the sentence appearseems to say that these genera only exist in the Iberian peninsula. Should appear : “The genera Oryctolagus and Lepus (order Lagomorpha) are an essential element of the trophic chain in Iberian Peninsula, being the main prey….”
Changed as proposed.
- Line 37 “L. granatensis is the widest distributed hare species in the Peninsula, and the only one present in the south of Spain (Palacios & Mejide 1979) and in Portugal”
Added as indicated. Sorry for this mistake.
- Lines 62 to 65 – This paragraph is not well written and is not scientific-like language. What is “veterinary criteria”?
We have modified this paragraph that is now as follows:
After several years collecting lagomorph data an epidemics outbreak occurred within hares, caused by MYXV-tol virus [13], ascribed to myxomatosis, according to external symptoms and veterinary analyses [14,15]. Dead or ill hares with myxomatosis symptoms were initially collected in the study area by local hunters. These samples were subsequently analyzed by the veterinary services of the Junta de Andalucía, that confirmed this disease and kept track of it in the area.
- Line 73: herbicide is a pesticide. Remove the first.
Done
- Line 77: rabbit hemorrhagic disease (RHD)
Changed
- Line 79: The authors should use official information and criteria, not subjective criteria. The lynx is endangered (IUCN) and not “highly endangered” and the Iberian imperial eagle is Vulnerable (IUCN).
Changed
- Figure 1 – The hare points are yellow, not red
Changed in caption
- Line 178 – Final point missing
Added
- Line 214 – “Could this reduction be attributed to the same infectious 214 outbreak observed in Córdoba?” Questions should not appear in the results. Remove.
We have reworded this point in the Results and moved to the discussion the reflections on dates and coincidence of the outbreak on both species.
- Line 380 “In some cases, the 380 local disappearance of rabbit populations due to myxomatosis were followed by increases 381 in hare populations (e.g. Britain) or not (e.g. Hungary). ”. Include the reference or remove.
We have added the following reference:
Sumption, K. J., & Flowerdew, J. R. (1985). The ecological effects of the decline in rabbits (Oryctolagus cuniculus L.) due to myxomatosis. Mammal Review, 15(4), 151-186.
- Line 411 – The authors should not talk about myxomatosis caused by classic MYXV and ha-MYXV like they knew that the diseases are the same and have the same characteristics. In fact, viruses behave completely differently (Evaluation of commercial vaccines (https://doi.org/10.3390/vaccines10030356), and it was known not long ago (Abade dos Santos, 2022. Evaluation of commercial vaccines…) that antibodies produced against the classic virus protect against the new ha-MYXV virus, which is naturally recombinant, but it is not certain that they protect during 18 months. Rephrase this paragraph accordingly.
Rephrased as follows:
The first clinically affected Iberian hares by myxomatosis were observed on 20 June 2018 in the province of Cuenca (central Spain) and new affected sites were progressively aggregated from southern and central regions (García Bocanegra et al, 2021). As the mean time interval between the first and maximum number of hares detected with myxomatosis was 31 days, the start of the epidemics in the Carmona’s countryside could be well before July 18, because this was the date when we stopped observing hares. The spatial distribution of the initial hare mortality was quite homogeneous throughout the affected areas in Cordoba´s province
- Figure 7 – Species name should appear italicized
Done
- Line 452 – “It is known that disease is a major source of deaths in hares (Edwards et al. 2000)” Which disease?
We want to mean: Diseases in general. We have corrected the sentence and verbal form
It is known that diseases are a major source of deaths in hares (Edwards et al. 2000). Hare populations are at their highest densities in autumn, with a high proportion of juveniles (Edwards et al. 2000).
- Line 472 – The same article referred before (Evaluation of commercial vaccines) proved that rabbits are highly susceptible to both natural recombinant virus (ha-MYXV) isolated from wild rabbit and from Iberian hare, so it is clear that transmission could appear between these two species. In fact, the transmission does not need to be direct, being much more likely an indirect transmission (insects, fleas...)
We have reworded it and referenced Abade Dos Santos
- Line 477 to line 480 – Remove the paragraph. It is well known that hares do not eat meat.
Done
- Line 481 to 483 – I can not understand what authors are hypothesizing. These species do cecotrophy but it not means that hare eats rabbit faeces. The process is the ingestion of the cecotrophs directly from the anus of the animal itself, not from the ground.
Yes, we known it. We want to refer that maybe animals could interspecifically smell excrements, for example, as an exploratory behaviour.
However, we acknowledge this is rather confusing, and we have eliminated it:
Our results show concordance with the hypothesis of blood-feeding insects, but the possibility of direct transmission by direct contact of hares with infected rabbit faeces cannot be excluded. This is mainly supported by the results obtained in the niche analysis: we found a similar spatial niche overlap, around 25%, between rabbits and hares in the study area prior and after the myxomatosis outbreak. Additionally, we found, as expected, that the niches of both species were not equivalent in the two scenarios, after and before the outbreak: were similar before the outbreak, but dissimilar after it.
- Line 490 – “This is com- 490 patible with the hypothesis that the remaining hares after the mortality are those avoiding 491 the contact with areas used by rabbits.”
This sentence is totally speculative. In fact, what happens is that animals do not like to compete for food and space, and hares are territorial animals. However, when there are high densities of both species it is normal for them to share the habitat, when there are few specimens it is normal for both species to retract to places with more food density, that is, not inhabited by other species.
Thank you very much for your very useful comment. It has made us to reanalyse the data, to intensely discuss about the spatial impact of the outbreak on rabbits and hares and, specially, on the meaning of the patterns observed. We think this is a key question of our work because it has important clues to understand the way the virus could be transmitted from rabbits to hares.
I’ll try to explain our discovery, based on a new way to present the results.
- We have divided de space defined by Factors 1 and 2 from the PCA into a mesh 0,5 points size. The locations visited, including those in which lagomorph were registered, are distributed in this space according to their X, Y, Z coordinates.
- In every square, we have computed the points visited (available) and the points with a lagomorph register, before and after July 2018 outbreak.
- We have calculated the frequency of lagomorph locations per square (hare or rabbit locations/total locations), before and after the outbreak.
- We have represented the frequency of each species, between species and within species, before and after the outbreak:
- Before the myxomatosis outbreak, rabbits and hares coexisted or not in the same area. After the outbreak, there was an almost complete exclusion of both species. It is pointing out that there was a direct transmission from one species to another. Presumably from rabbits to hares.
- There was a clear spatial segregation and a niche differentiation as a consequence of the outbreak.
- Our hypothesis is that myxomatosis is first originated in rabbits. It is then transmitted to hares coexisting with ill rabbits, via fleas/mosquitoes or some other alternative. Both, these hares and rabbits, die/disappear. Finally, only remain rabbits with an adequate immune response to myxoma virus and hares in sites with no contact with rabbits.
We have reworded this paragraph as follows:
Our results support the hypothesis of a direct transmission from rabbits to hares via fleas, blood-feeding insects or a similar mechanism. First, there was a clear spatial segregation of both species before and after the outbreak. Second, after the niche analysis we observed a higher niche similarity before the outbreak, but a lower one after it. According to the data presented here, the myxomatosis should have originated first in rabbits. It then was transmitted, via fleas/mosquitoes or some other alternative, to hares coexisting with ill rabbits. Both, these infected hares and rabbits die/disappear. As a consequence, only remained in the study area rabbits with an adequate immune response to myxoma virus and hares inhabiting in sites with no contact with rabbits.
- Line 509 – Lepus europaeus
Changed
- Figure 7 – It is not clear if authors are trying to represent a trophic chain. In any case, the direction of arrows is confused.
We have clarified it in caption, as follows:
Figure 8. Scheme of the impact of lagomorph depletion in the trophic chain of a pseudo-steppe agricultural land. The direction of arrows indicate which group predates or consumes over other.

Round 2
Reviewer 1 Report
New version of the manuscript has included my recommendations. Therefore, I consider that the manuscript can be published.